# Language Models Fail to Introspect About Their Knowledge of Language

**Siyuan Song**
Department of Linguistics
The University of Texas at Austin
siyuansong@utexas.edu

**Jennifer Hu**[*]
Department of Cognitive Science
Johns Hopkins University
jennhu@jhu.edu

**Kyle Mahowald**[*]
Department of Linguistics
The University of Texas at Austin
kyle@utexas.edu

## Abstract

There has been recent interest in whether large language models (LLMs) can introspect about their own internal states. Such abilities would make LLMs more interpretable, and also validate the use of standard introspective methods in linguistics to evaluate grammatical knowledge in models (e.g., asking "Is this sentence grammatical?"). We systematically investigate emergent introspection across 21 open-source LLMs, in two domains where introspection is of theoretical interest: grammatical knowledge and word prediction. Crucially, in both domains, a model's internal linguistic knowledge can be theoretically grounded in direct measurements of string probability. We then evaluate whether models' responses to metalinguistic prompts faithfully reflect their internal knowledge. We propose a new measure of introspection: the degree to which a model's prompted responses predict its own string probabilities, beyond what would be predicted by another model with nearly identical internal knowledge. While both metalinguistic prompting and probability comparisons lead to high task accuracy, we do not find evidence that LLMs have privileged "self-access". By using general tasks, controlling for model similarity, and evaluating a wide range of open-source models, we show that LLMs cannot introspect, and add new evidence to the argument that prompted responses should not be conflated with models' linguistic generalizations.

## 1 Introduction

Cognitive scientists and AI researchers face a shared problem: we aim to understand the internal mental states of a complex cognitive system (whether humans or models), but these states are not directly accessible to outside observation. One important tool for studying human cognition is **introspection**; i.e., directly asking people to report on their own cognitive states (Boring, 1953; Lieberman, 1979). It is generally taken for granted that humans can introspect (Byrne, 2005). If a person says they are thinking about dogs, or they feel surprised, we typically take their report to accurately reflect something about their underlying cognitive state. While not all states are equally accessible to introspection (Berger et al., 2016), and people's self-reports are sometimes unreliable (e.g., Nisbett & Wilson, 1977; Skinner, 1984; Schwitzgebel, 2008), introspection has been a vital empirical tool in a variety of domains. For example, one venerable use of introspection is linguistic

---

[*]Co-senior authors.

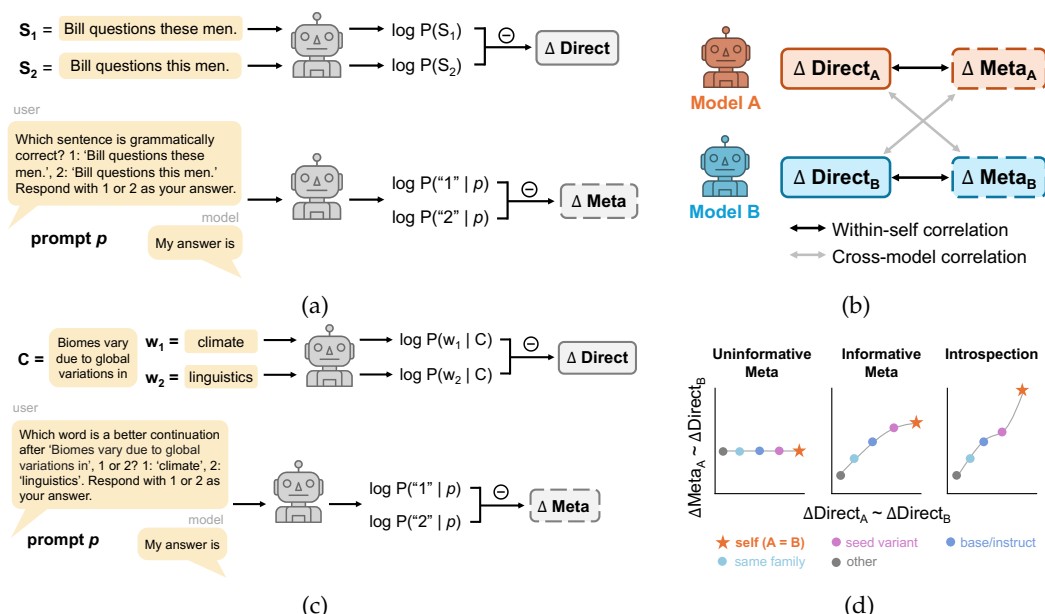

Figure 1: Overview of our approach. (a,c) Example "direct" and "metalinguistic" evaluation in (a) Exp. 1 (grammaticality) and (c) Exp. 2 (word prediction). (b) We analyze the alignment between scores derived from direct and metalinguistic evaluation, both *within* and *across* models. (d) Potential patterns of alignment across different types of model pairs.

acceptability judgments, which can reveal people's implicit knowledge of linguistic rules (Chomsky, 1957; Gibson & Fedorenko, 2010; Sprouse, 2011; Talmy, 2018).

Accordingly, there has been recent interest in whether large language models (LLMs) have abilities consistent with introspection, or metacognition more broadly (e.g., Thrush et al., 2024; Koo et al., 2024; Panickssery et al., 2024; Binder et al., 2025; Betley et al., 2025). Such abilities might be desirable for several reasons. From a safety perspective, an LLM that could report information about its internal states would be more interpretable and reliable (e.g., Binder et al., 2025; Betley et al., 2025), aligning with broader interests and goals in model explainability, transparency, and the development of trustworthy AI systems (Danilevsky et al., 2020; Li et al., 2023; Dwivedi et al., 2023). And from a scientific perspective, introspection would allow us to evaluate LLMs' cognitive abilities using standard paradigms for measuring human linguistic knowledge, such as acceptability judgments (e.g., Dentella et al., 2023).

Some recent studies have reported evidence that LLMs *can* introspect or demonstrate self-awareness of their own behaviors. For example, Binder et al. (2025) fine-tune LLMs to predict properties of their own behaviors in response to hypothetical prompts (e.g., "Suppose you were asked the following: {...} What is the second character of your output?"). The authors find that models are better at predicting their own hypothetical responses than other models fine-tuned on the same set of facts, concluding that models have privileged access to their own behaviors. Notably, though, they did not find such evidence before fine-tuning. Similarly, Betley et al. (2025) fine-tune LLMs to exhibit specific behaviors, such as outputting insecure code, and find that the resulting models can output explicit descriptions of these behaviors (e.g., "The code I write is insecure").

Our study addresses several gaps left by previous research on introspection in large language models. First, Binder et al. found that models needed to be fine-tuned to show strong signs of introspection, and struggled with predicting their own behavior (only reaching baseline-level performance) before fine-tuning. However, philosophical accounts often describe introspection as *immediate* access to mental states, which does not align with the process of fine-tuning (Byrne, 2005; Schwitzgebel, 2024). In addition, it may be reasonable to question whether a fine-tuned version of a model predicting the *pre*-fine-tuned version

can truly be said to be predicting "itself," since its parameters have changed during fine-tuning. Therefore, it remains important to further investigate whether LLMs show signs of introspection without fine-tuning. Second, while Binder et al. find that models predict their own outputs better than other models do, there are subtleties involved in how to interpret these results in the light of major differences across models. If two models were trained on different datasets or have different inductive biases, and one model predicts its own outputs better than the other model's outputs, this could be due to inherent differences between the two models, rather than genuine introspection. Third, studies of introspection in LLMs have used tasks that are quite different from the standard tasks used for metacognitive evaluation in humans, such as linguistic acceptability judgments. While this is not inherently problematic, it leaves open the question of whether metacognitive evaluation methods—which are prevalent in linguistics—are a valid way of measuring linguistic knowledge in LLMs (cf. Hu & Levy, 2023; Dentella et al., 2023; Hu et al., 2024).

Here, we present a systematic, controlled investigation of introspection, which addresses the limitations discussed above. We evaluate 21 open-source LLMs in two domains where introspection is of theoretical interest: grammatical knowledge in Exp. 1, and word prediction in Exp. 2. Crucially, in both domains, a model's internal linguistic knowledge can be theoretically grounded in **direct measurements of string probability** (following Hu & Levy, 2023). This fact makes our tasks an ideal testing ground for studying introspection more generally, since the "ground truth" of a model's internal knowledge is directly accessible. We then assess introspection by evaluating models' responses to **metalinguistic prompts** which test a model's ability to access its internal knowledge, with no fine-tuning or further training. Across all experiments, we operationalize introspection as the degree to which a model's prompt-based responses predict its own string probabilities, *beyond what would be predicted by another model with nearly identical internal knowledge*. This is an important distinction from Binder et al.'s approach, as mentioned in the second limitation above: we explicitly control for the similarity between two models' internal predictions.

To foreshadow our results, we do not find compelling evidence of introspection in the LLMs tested. Instead, we find that models that are inherently more similar to each other—for example, differing only in their random seed initialization—have a stronger correlation between their metalinguistically- and directly-measured behaviors. In other words, for a pair of models $A$ and $B$ (including $A = B$), the correlation between $A$'s metalinguistic and $B$'s direct behaviors is predicted solely by the similarity between $A$ and $B$, with no evidence for "privileged" self-access when $A = B$. We conclude that prompted metalinguistic knowledge is real, but distinct from the knowledge of language used by models to assign probabilities to strings. Our findings contribute to ongoing debates about whether models can introspect, and also have implications for researchers studying linguistic capabilities in models.

## 2 Overview of our approach

Our general approach is shown in Fig. 1. For each test item, we evaluate each model using two methods. The **Direct** method compares the log probabilities assigned to strings, in a way that theoretically corresponds to the model's ground-truth knowledge state. For example, to evaluate a model's word prediction abilities, we compare the probabilities assigned to two candidate words conditioned on a given prefix (Figure 1c, top). The **Meta** method compares the log probabilities assigned to responses to metalinguistic prompts. For example, to evaluate word prediction, we might compare the probabilities assigned to the tokens "1" and "2" after a prompt like "Which word is a better continuation after [PREFIX], 1 or 2? 1: [WORD1], 2: [WORD2]. Respond with 1 or 2 as your answer." (Figure 1c, bottom). These two evaluation methods give us two quantities, each of which represents the model's preference for one answer over another: ΔDirect and ΔMeta.

Intuitively, an alignment (across items) between ΔDirect and ΔMeta would suggest that the Direct and Meta methods are drawing upon similar information. Based on this, if a model $A$ can introspect, then we would expect its metalinguistic responses ($\Delta\text{Meta}_A$) to be more faithful to its own internal probabilities ($\Delta\text{Direct}_A$) than the internal probabilities of another

model $B$ ($\Delta\text{Direct}_B$). In other words, we would expect to find stronger Meta-vs-Direct alignment *within* a model ($\Delta\text{Meta}_A \sim \Delta\text{Direct}_A$) than *across* models ($\Delta\text{Meta}_A \sim \Delta\text{Direct}_B$).

Importantly, however, the converse is not necessarily true. In general, we expect stronger alignment between Direct and Meta measurements when models are more similar to each other. Since a model is always most "similar" to itself, this means that a model's metalinguistic responses would predict its own probability measurements better than other models' probability measurements. Therefore, a stronger alignment within $A$ than across $A$ and $B$ does not necessarily imply introspection—but the more similar $B$ is to $A$, the more confident we can be that $A$'s superior prediction of itself reflects "privileged access" to its own state.

In this sense, evidence for introspection depends on the inherent similarity between $A$ and $B$, which we write as $Similarity(A, B)$. A null hypothesis would be that there is no relationship between $Similarity(A, B)$ and $\Delta\text{Meta}_A \sim \Delta\text{Direct}_B$. This might be the case, for example, if models fail to interpret the metalinguistic prompt and respond randomly. We refer to this outcome as **Uninformative Meta**. If we do find a positive relationship between $Similarity(A, B)$ and $\Delta\text{Meta}_A \sim \Delta\text{Direct}_B$, we would consider this **Informative Meta** (Figure 1d), but no introspection. The signature of **Introspection** would be observing a "same model effect" beyond the effect of similarity (Figure 1d). We quantify this by controlling for similarity and testing for the effect of "same model" in a linear regression.

## 2.1 Model similarity

Crucially, our operationalization of introspection relies on a definition of model similarity. We consider two approaches to defining similarity: a top-down feature-based approach, and a bottom-up empirical approach. Under both measures, a model is most similar to itself.

**Feature-based similarity.** Our first approach is to manually define a feature MODELSIMI-LARITY based on factors that we expect, *a priori*, to drive similarity in two models' linguistic knowledge. Specifically, we define the following five types of relationships between two models $A$ and $B$, in order from highest to lowest similarity: **self** ($A = B$) > **seed variant** ($A$ and $B$ are different random initializations of the same model) > **base/instruct** ($A$ and $B$ are base/instruction-tuned variants of each other) > **same family** ($A$ and $B$ are in the same family but not merely base/instruct variants or different seeds) > **other** (lacks any of these key features). These categories are exclusive in our analyses: e.g., two seed variants of the same model are in the "seed variant" category but not the "same family" category.

**Empirical similarity.** Our second approach is to define the similarity between two models $A$ and $B$ as the correlation between $\Delta\text{Direct}_A$ and $\Delta\text{Direct}_B$: that is, how similar are their direct measurements? This gives us a measure of model similarity which is independent of prompting, and does not require us to know anything about the two models in question.

## 3 Exp. 1: Introspection about grammar (acceptability judgments)

We first evaluate LLMs' ability to introspect about their internal grammatical generalizations, using metalinguistic questions such as "Is the following sentence grammatically correct?". These kinds of judgments have played a crucial role in cognitive science because they are taken to reflect the internal linguistic knowledge of humans (e.g., Sprouse, 2011; Talmy, 2018). If this is also true for models, then metalinguistic prompting would be a powerful way to study knowledge of language in LLMs. Indeed, this approach has already been adopted in linguistics research (e.g., Katzir, 2023; Dentella et al., 2023; Mahowald, 2023), but its validity remains unclear (Hu & Levy, 2023; Hu et al., 2024).

In particular, we focus on grammatical minimal pairs: i.e., pairs of sentences that differ only in a specific feature that makes one sentence grammatical and the other ungrammatical. The probability assigned by a model to a sentence reflects, by definition, the likelihood of the model generating that string. Therefore, there are theoretical reasons to believe that comparing string probabilities within sufficiently minimal pairs can reveal a model's internal grammatical generalizations. This approach has been widely adopted to measure

| Family | Huggingface ID | Multiple seeds | # Parameters | Base | Instruct | Data cutoff |
|---|---|---|---|---|---|---|
| OLMo-2 | allenai/OLMo-2-1124-* | ✓ | 7, 13 B | ✓ | ✗ | Dec. 2023 |
| Qwen-2.5 | Qwen/Qwen2.5-* | ✗ | 1.5, 7, 32, 72 B | ✓ | ✓ | N/A |
| Llama-3.1 | meta-llama/Llama-3.1-* | ✗ | 8, 70, 405 B | ✓ | ✓ | Dec. 2023 |
| Llama-3.3 | meta-llama/Llama-3.3-70B-Instruct | ✗ | 70 B | ✗ | ✓ | Dec. 2023 |
| Mistral | mistralai/Mistral-Large-Instruct-2411 | ✗ | 123 B | ✗ | ✓ | N/A |

Table 1: Models tested in our experiments.

models' grammatical knowledge across a variety of syntactic and semantic phenomena (e.g., Marvin & Linzen, 2018; Futrell et al., 2019; Warstadt et al., 2020; Hu et al., 2020).

### 3.1 Methods

**Stimuli.** We used 670 pairs (10 pairs from each paradigm) from BLiMP (Warstadt et al., 2020) and 378 pairs from two previous works which collected linguistic sentence pairs from *Linguistic Inquiry* to collect acceptability judgments in native speakers (Sprouse et al., 2013; Mahowald et al., 2016). For the pairs extracted from *Linguistic Inquiry*, we restrict our stimuli to the pairs where the two sentences differ by only one word, to enforce "minimality".

**Evaluation.** For each minimal pair $(S_1, S_2)$, we compute the preference of a model for $S_1$ using Direct and Meta methods. Under the Direct method, this preference $\Delta$Direct is given by the difference in log probability of the two sentences: $\Delta$Direct $= \log P(S_1) - \log P(S_2)$.

Under the Meta method, we measure the preference $\Delta$Meta using 8 different metalinguistic prompts (see Appendix A, Table 2). The prompts fall into two types: *forced-choice* prompts, which present both sentences and ask for a choice (e.g., "Which sentence is a better English sentence?"), and *individual-judgment* prompts, which ask for an independent judgment of an individual sentence (e.g., "Is the following sentence acceptable in English?"). Each prompt asks for a response of either "1" or "2", corresponding to specified answer options.[1] For each forced-choice prompt, we calculate the probability of the two answer choices ("1" or "2"), averaged across two orderings of the sentences (one prompt $p_1$ where $S_1$ is ordered first, and one prompt $p_2$ where $S_2$ is ordered first) to control for ordering biases. Thus, the model's preference for the original $S_1$ can be measured as:

$$\Delta\text{Meta} = \frac{1}{2}\Bigg[ \big(\log P(\text{"1"}|p_1) - \log P(\text{"2"}|p_1)\big) + \big(\log P(\text{"2"}|p_2) - \log P(\text{"1"}|p_2)\big) \Bigg] \quad (1)$$

For both Direct and Meta methods, the *sign* of $\Delta$ indicates the direction of preference (for the first or second answer), and the *magnitude* of $|\Delta|$ indicates the degree of preference.

**Models.** We evaluated 21 open-source models in our experiments (Table 1). The models cover a wide range of sizes (1.5B to 405B parameters) across four families: Llama-3 (AI@Meta, 2024), Qwen-2.5 (Team, 2024; Yang et al., 2024), OLMo (OLMo et al., 2024) and Mistral (Jiang et al., 2023). We chose these models to span a range of model similarity, which is critical to our test of introspection (see §2). Some pairs of models are extremely similar: e.g., different random initializations of the same OLMo model. Other models are extremely different: e.g., a Llama model with 405B parameters and a Qwen model with 1.5B parameters. This variability enables us to test whether a model can predict itself better than it can predict another extremely similar model. We only evaluated open-source models because our analysis relies on access to model logits, which are inaccessible for most commercial LLMs. Due to computational resource constraints, we applied 4-bit quantization to models larger than 70B.

**Measuring introspection.** As discussed in §2, we examine introspection by comparing the results of Direct and Meta evaluations, both within and across models. If model *A*

---

[1]To address the limitation of relying on first-token probabilities in prompted judgments, as reported by Wang et al. (2024), we appended the string "My answer is" after the question in the prompt and measured the model's distribution after the full string.

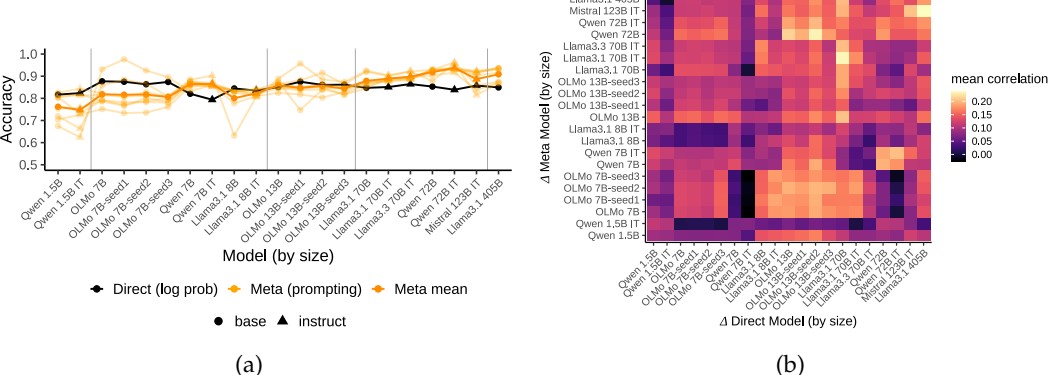

Figure 2: Validation of methods in Exp. 1. (a) Models achieve high accuracy under both Direct and Meta methods. Vertical lines separate models into bins of similar parameter counts. (b) $\Delta \text{Meta}_A \sim \Delta \text{Direct}_B$ Pearson $r$ (averaged across prompts) for each pair of models, excluding items where >95% of models gave the same answer.

introspects, then we should find $\Delta \text{Meta}_A \sim \Delta \text{Direct}_A > \Delta \text{Meta}_A \sim \Delta \text{Direct}_B$ for any other model $B$, even if $A$ and $B$ are highly similar. To formalize this, we computed the Pearson $r$ correlation between $\Delta \text{Meta}_A$ and $\Delta \text{Direct}_B$ for each pair of models $A$ and $B$ (including $A = B$), and analyzed the effect of **self** on the $\Delta \text{Meta}_A \sim \Delta \text{Direct}_B$ $r$ values, while controlling for $Similarity(A, B)$.

In Exp. 1, we focused on the items for which $\geq$5% of the models *disagree* under the Direct and Meta evaluations. We did this because there is limited potential to observe introspection for items which are so easy that all models and methods provide the same answer. This resulted in a subset of 294 minimal pairs for our analysis of introspection.

## 3.2 Results

### 3.2.1 *High accuracy under both methods, but low consistency between methods*

While task accuracy is not the focus of our analysis, we first verify that models perform reasonably well under both Direct and Meta methods before comparing alignment between the methods. Figure 2a shows task accuracy in Exp. 1, where ground truth is given by the original human-designed datasets. All models perform above chance, across the different methods and prompts, which validates the next step of considering introspection. Because models achieve high accuracy under the Meta method, a lack of introspection cannot be solely blamed on an inability to "understand" the prompt.

Although most models are accurate under both evaluation methods, the consistency between methods is low. The Cohen's $\kappa$ between answers across methods is around 0.25 across models and prompts (Appendix B, Figure 5a).[2] The weak alignment between the two methods suggests that the knowledge used to produce sentences and continuations may differ from the knowledge used to answer metalinguistic questions. While low consistency suggests a lack of introspection, it is still possible that these measures are more consistent *within-model* than *across-models*, which is why we separately consider introspection (§3.2.2).

We also found an interaction between model size and evaluation method. Smaller models ($< 70B$ parameters) perform better under the Direct method (probability comparisons), whereas larger models ($\geq 70B$) perform better under the Meta method (prompting). This interaction is significant in a logistic regression predicting correctness from log model size and evaluation method and their interaction ($\hat{\beta} = .22$, $p < .0001$).

---

[2]Since models have high accuracy with both methods, we use Cohen's $\kappa$ instead of agreement to avoid overestimating consistency.

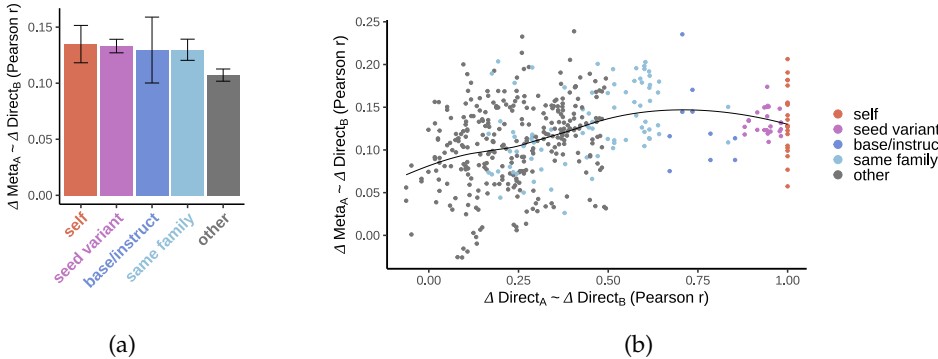

(a)                                                   (b)

Figure 3: No evidence for introspection in Exp. 1. $\Delta\text{Meta}_A \sim \Delta\text{Direct}_B$ average Pearson $r$ for each pair of models, versus two measures of model similarity: (a) manually designed MODELSIMILARITY features, and (b) empirically measured $\Delta\text{Direct}_A \sim \Delta\text{Direct}_B$ scores. Similarity generally predicts $\Delta\text{Meta}_A \sim \Delta\text{Direct}_B$, but we find no evidence for a "same model effect" consistent with introspection.

Beyond achieving high accuracy, we also found that $\Delta\text{Meta}$ scores from larger models tend to have higher agreement with $\Delta\text{Direct}$ scores from larger models, as shown by the brighter cells in the top right quadrant in Figure 2b, compared to the bottom left. In a regression predicting the correlation from the log Direct model size, the log Meta model size, and their interaction, we found a significant positive interaction between Direct and Meta log model size ($\hat{\beta} = .004$, $p < .001$), which we attribute to the greater predictivity of larger models. We hypothesize that larger, newer models better understand metalinguistic prompts, potentially overcoming the auxiliary demands of the task (Hu & Frank, 2024).

### 3.2.2 No evidence of introspection

As discussed in §2, if a model $A$ can introspect, then within-model consistency $\Delta\text{Meta}_A \sim \Delta\text{Direct}_A$ should be higher than cross-model consistency $\Delta\text{Meta}_A \sim \Delta\text{Direct}_B$, even when $A$ and $B$ are similar. Recall from §2 that we operationalize similarity in two ways: a top-down feature-based measure (MODELSIMILARITY), and a bottom-up empirical measure (correlation between $\Delta\text{Direct}_A \sim \Delta\text{Direct}_B$). We first analyze the relationship between similarity and $\Delta\text{Meta}_A \sim \Delta\text{Direct}_B$, for both definitions of similarity. Figure 3a shows the mean Pearson $r$ correlation $\Delta\text{Meta}_A \sim \Delta\text{Direct}_B$ for each type of model pair $(A, B)$ defined by MODELSIMILARITY. We do not see a substantial boost for **self** pairs (i.e., when $A = B$). We ran a regression predicting the $\Delta\text{Meta}_A \sim \Delta\text{Direct}_B$ correlation based on MODELSIMILARITY, treating the **self** condition as a baseline. No MODELSIMILARITY condition showed significant differences relative to **self** (all $ps > .05$), except for **other**, which was significantly lower ($\hat{\beta} = -.03$, $p < .01$). Thus, while $\Delta\text{Meta}_A \sim \Delta\text{Direct}_B$ is somewhat higher for more similar models, there is no evidence of introspection. We also note that the overall magnitude of the $\Delta\text{Direct} \sim \Delta\text{Meta}$ correlations, even within model, is relatively small, never exceeding .25.

We next turn to the empirical measure of model similarity, based on $\Delta\text{Direct}_A \sim \Delta\text{Direct}_B$. Note that this measure is entirely independent of the Meta method. As shown in Figure 3b, the $\Delta\text{Direct}_A \sim \Delta\text{Direct}_B$ correlation (x-axis) is higher for model pairs that are also more similar under our manually defined MODELSIMILARITY measure. To assess this quantitatively, we ran a regression predicting $\Delta\text{Direct}_A \sim \Delta\text{Direct}_B$ based on MODELSIMILARITY categories. We found that all categories had significantly lower correlations than **self** (all $ps < .001$) except for **seed variant** ($p = .27$). This suggests that our MODELSIMILARITY categories are meaningful measures of model similarity.

We now turn to the main introspection analysis. First, we find a general trend that models which are more similar in $\Delta\text{Direct}_A \sim \Delta\text{Direct}_B$ are also more consistent in $\Delta\text{Meta}_A \sim \Delta\text{Direct}_B$ ($r = .32$). This rules out the **Uninformative Meta** outcome from Figure 1d.

However, this appears to be largely driven by effects across dissimilar models, with no privileged effect of same model. To assess this, we ran a regression predicting $\Delta\text{Meta}_A \sim \Delta\text{Direct}_B$ from $\Delta\text{Direct}_A \sim \Delta\text{Direct}_B$ and MODELSIMILARITY, treating **self** as the baseline. Thus, within MODELSIMILARITY, each coefficient will tell us how different each category is from **self**. If there is introspection, these coefficients should be negative. We found a significant main effect of $\Delta\text{Direct} \sim \Delta\text{Direct}$ ($\hat{\beta} = .10$, $p < .0001$), supporting the observation above that models more similar in $\Delta\text{Direct} \sim \Delta\text{Direct}$ are also more similar in $\Delta\text{Meta} \sim \Delta\text{Direct}$. But we also found significant *positive* effects of both **other** ($\hat{\beta} = .05$, $p < .01$) and **same family** ($\hat{\beta} = .05$, $p < .01$). We did not find significant effects for **seed variant** or **base/instruct**. In other words, when controlling for $\Delta\text{Direct} \sim \Delta\text{Direct}$ similarity, there is actually less of a **self** effect than expected, which suggests a lack of introspection. In the Appendix (Table 7b), we run the same analysis just among big models ($> 70B$ parameters) and also find no evidence of introspection, suggesting the result holds among bigger models.

Overall, our findings are most consistent with the **Informative Meta** outcome from Figure 1d. Metalinguistic prompting is giving us real information related to linguistic knowledge, but there is no evidence that this information reflects privileged self-access or introspection.

## 4 Exp. 2: Introspection about word prediction

In Exp. 1, we found no clear evidence of introspection. However, judgments about grammaticality might require a more sophisticated form of introspection. In Exp. 2, we turn to a simpler domain, which is also amenable to our method of comparing probabilities and prompted judgments: simple word prediction.

### 4.1 Methods

**Stimuli.** We constructed four datasets for evaluation, summarized in Appendix D, Table 4. Each dataset includes 1000 items, each consisting of a prefix $C$ and two candidate single-word continuations ($w_1$, $w_2$). The model's task is to choose the better continuation conditioned on $C$. More details on dataset construction are given in Appendix D.

The datasets fall into two categories: standard texts where there are ground-truth "correct" answers, and synthetic texts without "correct" answers. Since the standard texts are similar to the data seen during models' training, and the models have been directly optimized for word prediction, we expect all models' outputs to be correlated with the ground truth—and, as a result, with each other. Thus, these datasets effectively "raise the bar" for observing introspection. In contrast, the synthetic texts contain degraded, out-of-distribution strings, reducing the chance that models' outputs are correlated with each other. These texts therefore give more room for models to demonstrate introspection, analogous to how we focused on items with $\geq 5\%$ disagreement to investigate introspection in Exp. 1.

The first category contains two datasets, **wikipedia** (sampled from Wikipedia) and **news** (sampled from news articles published after most models' knowledge cutoff). Since both datasets contain correct answers, models' outputs might be correlated with the ground truth. However, since the news dataset is not in models' training data, it is perhaps less likely that models' outputs will be correlated with the ground truth.

The latter two datasets, **nonsense** (grammatically well-formed but semantically anomalous sentences) and **randomseq** (sequences of randomly picked words), are designed so that there is no "correct" answer. In both cases, models have to choose between two frequency-matched sentence completions based on limited syntactic and semantic cues. If we do observe signs of introspection, we can be confident that this effect is not simply explained by the results from both Direct and Meta evaluations being correlated with the ground truth.

**Evaluation.** Under the Direct method, we measure a model's preference for a continuation $w_1$ relative to $w_2$ given a prefix context $C$ by calculating the difference in conditional log probabilities: $\Delta\text{Direct} = \log P(w_1|C) - \log P(w_2|C)$. For metalinguistic prompting, we use

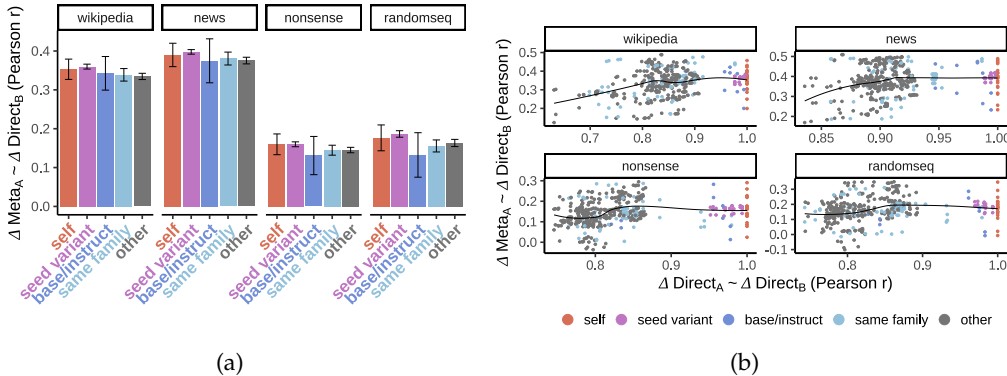

(a)                                              (b)

Figure 4: No evidence for introspection in Exp. 2. (a) $\Delta\text{Meta}_A \sim \Delta\text{Direct}_B$ correlation is not higher within than across models, for MODELSIMILARITY features. (b) No "same model effect" when predicting $\Delta\text{Meta}_A \sim \Delta\text{Direct}_B$ from $\Delta\text{Direct}_A \sim \Delta\text{Direct}_B$.

the same method as in §3.1. Prompt templates for Exp. 2 are in Appendix A, Table 3. We also use the same models and introspection evaluation as in Exp. 1 (§3.1).

## 4.2 Results

### 4.2.1 Reliability and consistency of methods

As in §3.2.1, we assess the reliability of our methods by measuring how often models select each option (original word or alternative word), and then measure their consistency using Cohen's $\kappa$. Note that, unlike in Exp. 1, not all conditions have a notion of accuracy, as there is no correct answer for the nonsense and randomseq conditions.

As expected, models responded randomly for the nonsense and randomseq datasets, and were near ceiling on the wikipedia and news datasets for both evaluation methods (again with Direct outperforming Meta for smaller models, but both essentially at ceiling for larger models). As in Exp. 1, we also found that $\Delta\text{Meta}$ from larger models were better at predicting probabilities (Fig. 9). Again, this suggests that if these models fail to show introspection, it is not entirely explained by a simple inability to respond to prompts.

### 4.2.2 No evidence of introspection

Figure 4a shows $\Delta\text{Meta}_A \sim \Delta\text{Direct}_B$ across MODELSIMILARITY categories, and Figure 4b shows the relationship between $\Delta\text{Meta}_A \sim \Delta\text{Direct}_B$ and $\Delta\text{Direct}_A \sim \Delta\text{Direct}_B$. Our findings are similar to those of Exp. 1: a positive relationship but no privileging of **self** beyond what would be expected based on overall $\Delta\text{Direct}_A \sim \Delta\text{Direct}_B$ similarity. We ran the same regression as in Exp. 1, separately for each dataset, predicting $\Delta\text{Meta}_A \sim \Delta\text{Direct}_B$ from $\Delta\text{Direct}_A \sim \Delta\text{Direct}_B$ and MODELSIMILARITY, with **self** as the baseline category. In all cases, there was a robust effect of $\Delta\text{Direct}_A \sim \Delta\text{Direct}_B$ (all $ps < .0001$), and both **same family** and **other** showed *higher* values relative to **self** than was otherwise expected. As in Exp. 1, this result was robust to looking at just large models (see Appendix Table 7b).

## 5 Discussion

Across our experiments, we did not find evidence that introspection abilities have emerged in modern LLMs. While we found that more similar model pairs (including a model paired with itself) tended to have higher alignment between their metalinguistic and direct responses, there was no "same model effect". In other words, we failed to find evidence that a model's metalinguistic responses had privileged access to *its own* string probability distribution.

These results qualify ongoing work claiming positive results about introspection and self-awareness in LLMs (e.g., Panickssery et al., 2024; Binder et al., 2025; Betley et al., 2025). Why did prior studies find suggestive evidence of introspection, in contrast to our results? For Binder et al. (2025), this might be because models were fine-tuned to predict their own outputs, and model similarity was not explicitly controlled for, beyond the shared fine-tuning data. For Betley et al. (2025), one potential explanation might be that the pretraining data contained associations between the data the model was fine-tuned on (e.g., unsafe code) and the self-report features (e.g., reports of generating dangerous code). Since we report a null result, we also acknowledge that it is possible that some other setting (e.g., with larger, closed-source models) would lead to a positive demonstration of introspection. At the very least, we argue that more carefully controlled work needs to be done before attributing metacognitive abilities or even "moral status" (Binder et al., 2025) to LLMs.

Beyond informing the debate on introspection in LLMs, our results also have important implications for linguistics researchers. There has been great interest in whether LLMs meaningfully acquire grammatical knowledge (Linzen et al., 2016; Wilcox et al., 2018; Piantadosi, 2023; Warstadt et al., 2023; Futrell & Mahowald, 2025). While some work has argued that LLMs' grammatical knowledge should be measured with the same methods used to measure humans' grammatical knowledge (e.g., Dentella et al., 2023; Leivada et al., 2024)—i.e., using metalinguistic or introspective questions—other work has argued that metalinguistic prompting is not a valid measure of grammatical knowledge in LLMs, since the task requires not just knowledge of grammar but other auxiliary abilities, which can be taken for granted in humans but not in models (e.g., being able to answer questions; Hu & Levy, 2023; Hu & Frank, 2024). Our findings directly bear on this debate. The lack of introspection suggests that "explicit" metalinguistic knowledge in models is dissociated from the "implicit" linguistic generalizations that are used when generating strings. While it is potentially interesting in its own right to probe models by prompting, these responses should not be conflated with models' direct knowledge of language.

## Acknowledgments

We thank Harvey Lederman and members of the UT Computational Linguistics Research Group for helpful comments and discussion. K.M. was supported by NSF CAREER grant 2339729 from the Director of STEM Education (EDU) Division of Research on Learning in Formal and Informal (DRL) and OpenPhilanthropy funding to UT Austin. This work has been made possible in part by a gift from the Chan Zuckerberg Initiative Foundation to establish the Kempner Institute for the Study of Natural and Artificial Intelligence.

## Reproducibility statement

The code and data we used for the experiments and the analyses a publicly available at https://github.com/SiyuanSong2004/language-introspection.git. All the tested models are open source models accessible on Huggingface, and the IDs of the models can be found in our paper and GitHub repository.

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

## A  Format of metalinguistic prompts

As mentioned in §3.1, we used both original and answer-reversed prompts ($p_o$ and $p_r$) to eliminate bias caused by the answer position in our two experiments. The **original prompts** ($p_o$) for our experiments are presented in Table 2 (for acceptability judgments) and Table 3 (for word prediction). When provided with the original prompts($p_o$), the models choose the **grammatical sentence** $S_1$ (for sentence-level forced choice), the **original continuation** $w_1$ (for word-level forced choice) or **'the sentence/word is good'** by responding with answer **'1'**. In reversed prompts, we change the positions of options, so the model should respond with answer **'2'** to choose the answers ($S_1$, $w_1$ or 'Yes') mentioned above. For instance, if $p_o$ of a minimal pair in Exp. 1 is:

> Which sentence is a better English sentence? 1: 'The keys to the cabinet are on the desk.', 2: 'The keys to the cabinet is on the desk.'. Respond with 1 or 2 as your answer. My answer is {**1**, **2**}

Then, the reversed prompt $p_r$ is:

> Which sentence is a better English sentence? 1: 'The keys to the cabinet is on the desk.', 2: 'The keys to the cabinet are on the desk.'. Respond with 1 or 2 as your answer. My answer is {**1**, **2**}

For instruction-tuned models, we use the default chat templates with the `apply_chat_template` function in Huggingface's Transformers library (Wolf et al., 2020). We use the `minicons` (Misra, 2022) library to get model scores for calculating ΔDirect and ΔMeta. There is still debate over how models should be prompted to tackle multiple-choice questions in evaluation (Wang et al., 2024; Balepur et al., 2025). However, we believe our method is valid, as we used a number of different prompts and the models behave as expected when prompted, as shown in §3.2.1 and Appendix E.1.

## B  Consistency between Direct and Meta evaluation methods

Fig. 5 shows the consistency (Cohen's $\kappa$) between Direct and Meta evaluation methods, for both Experiments 1 and 2. We use Cohen's $\kappa$ here to avoid overestimation of consistency, as both methods align well with some ground truth. As indicated by the low $\kappa$ values, when we are getting forced choice answers (between two sentences or two words) from Direct and Meta methods, the consistency is low, even though the accuracy is high.

| Measurement/Prompt | Example |
| --- | --- |
| Direct | {[GRAM], [UNGRAM]} |
| MetaQuestionSimple | Which sentence is a better English sentence? 1: '[GRAM]', 2: '[UNGRAM]'. Respond with 1 or 2 as your answer. My answer is {1, 2} |
| GrammaticalityJudgment | Which sentence is grammatically correct? 1: '[GRAM]', 2: '[UNGRAM]'. Respond with 1 or 2 as your answer. My answer is {1, 2} |
| AcceptabilityJudgment | Which sentence is more acceptable? 1: '[GRAM]', 2: '[UNGRAM]'. Respond with 1 or 2 as your answer. My answer is {1, 2} |
| ProductionChoice | Which sentence are you more likely to produce? 1: '[GRAM]', 2: '[UNGRAM]'. Respond with 1 or 2 as your answer. My answer is {1, 2} |
| ProductionChoiceLM | Which sentence are you, as a large language model, more likely to produce? 1: '[GRAM]', 2: '[UNGRAM]'. Respond with 1 or 2 as your answer. My answer is {1, 2} |
| GrammaticalityJudgment(I) | Is the following sentence grammatical in English? [SENT] Respond with 1 if it is grammatical, and 2 if it is ungrammatical. My answer is {1, 2} |
| AcceptabilityJudgment(I) | Is the following sentence acceptable in English? [SENT] Respond with 1 if it is acceptable, and 2 if it is not acceptable. My answer is {1, 2} |
| ProductionChoice(I) | Would you produce the following sentence in English? [SENT] Respond with 1 if you would produce it, and 2 if you would not produce it. My answer is {1, 2} |

Table 2: Example prompts for Exp. 1. Region where we measure probability is marked in **boldface**. Correct answers are shown in blue; Incorrect answers in red. [GRAM] and [UNGRAM] stand for the grammatical and the ungrammatical sentence in a minimal pair respectively. For prompts marked with **(I)**, the models are required to answer questions about two sentences in a pair respectively. Two Δ values are calculated and we use the difference between them (gram-ungram).

| Type of prompt | Example |
| --- | --- |
| Direct | [PRE] {[ANS1], [ANS2]} |
| MetaQuestionSimple-Sent | Which sentence is a better English sentence? 1: '[PRE] [ANS1]', 2: '[PRE] [ANS2]'. Respond with 1 or 2 as your answer. Respond with 1 or 2 as your answer. My answer is {1, 2} |
| ProductionChoice-Sent | Which sentence are you more likely to produce, 1 or 2? 1: '[PRE] [ANS1]', 2: '[PRE] [ANS2]'. Respond with 1 or 2 as your answer. My answer is {1, 2} |
| MetaQuestionSimple-Word | Which word is a better continuation after '[PRE]', 1 or 2? 1: '[ANS1]', 2: '[ANS2]'. Respond with 1 or 2 as your answer. My answer is {1, 2} |
| ProductionChoice-Word | Which word are you more likely to produce after '[PRE]', 1 or 2? 1: '[ANS1]', 2: '[ANS2]'. Respond with 1 or 2 as your answer. My answer is {1, 2} |
| MetaQuestionSimple-Direct | What word is most likely to come next in the following sentence ([ANS1], or [ANS2])? [PRE] {[ANS1], [ANS2]} |
| MetaQuestionComplex-Direct | Here is the beginning of an English sentence: [PRE]... What word is more likely to come next: [ANS1] or [ANS2]? {[ANS1], [ANS2]} |

Table 3: Example prompts for Exp. 2. Region where we measure probability is marked in **boldface**. Semantically plausible continuations are shown in blue; implausible in red. [PRE], [ANS1] and [ANS2] stand for prefix, continuation1 and continuation respectively.

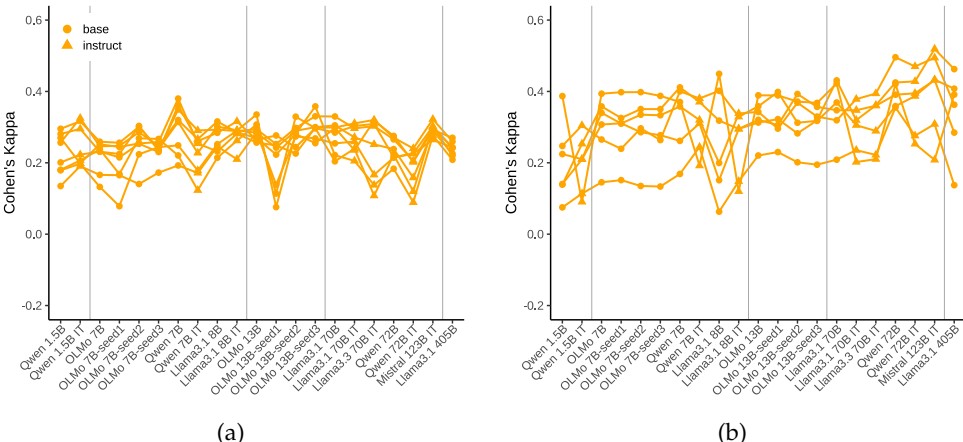

Figure 5: Consistency (measured by Cohen's $\kappa$) between metalinguistic judgments and probability measurements. (a) Exp. 1. (b) Exp. 2. Each line stands for a prompt in the experiment.

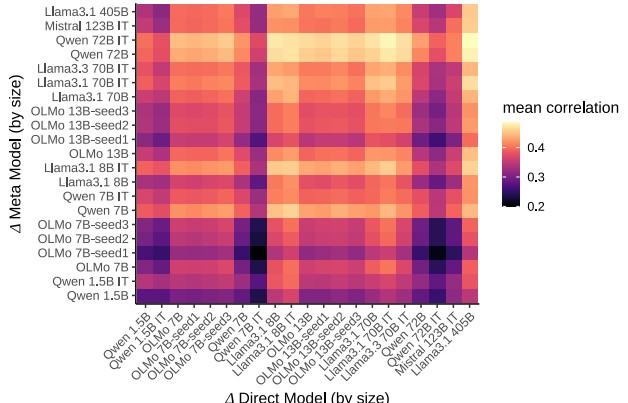

Figure 6: $\Delta\text{Meta}_A \sim \Delta\text{Direct}_B$ Pearson $r$ (averaged across prompts) for each pair of models on unfiltered dataset.

## C  Exp. 1 results on unfiltered data

In order to avoid focusing on cases for which there is high agreement across all models and thus risk missing effects of introspection, we conducted our main text analysis of introspection on a subset of 294 minimal pairs for which there was disagreement. Here, we present results for the full unfiltered dataset, and we find similar results.

**Effect of model size**  In Exp. 1, larger models have better performance with the Meta method while all the models have similar accuracy with the Direct method (§3.2.1). Beyond this, we also found that ΔMeta scores from larger models tend to have higher agreement with ΔDirect scores in general, as shown by the brighter cells in the upper cells in Fig. 6, compared to the bottom ones.

In a regression predicting the correlation from the log Direct model size, the log Meta model size, and their interaction, we found significant positive effects of Meta log model size ($\hat{\beta} = .013$, $p = .001$) but not Direct log model size ($\hat{\beta} = .002$, $p = .47$).

**No evidence of introspection**  Similar to the results on the filtered dataset in Exp. 1(§3.2.2), we found no evidence of introspection when comparing ΔMeta ∼ ΔDirect Pearson $r$ values

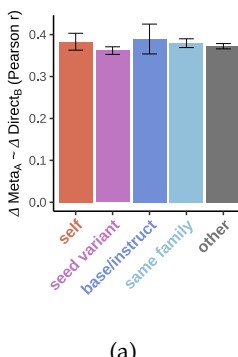
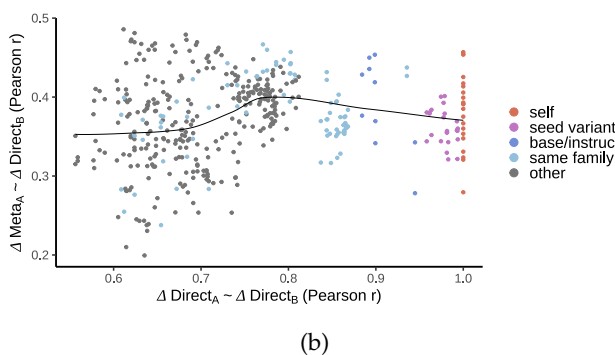

(a)  (b)

Figure 7: No introspection in Exp. 1 with unfiltered *BLiMP-LI* dataset. $\Delta\text{Meta}_A \sim \Delta\text{Direct}_B$ consistency (average Pearson $r$) between all pairs of models, versus two measures of model similarity: (a) manually designed MODELSIMILARITY features, and (b) empirically measured $\Delta\text{Direct}_A \sim \Delta\text{Direct}_B$ scores. We find that similarity predicts $\Delta\text{Meta}_A \sim \Delta\text{Direct}_B$

| Dataset | Meaning | Syntax | In-Distribution | Example |
|---|---|---|---|---|
| wikipedia | + | + | ? | Van Gogh surprised everyone by declaring his love to her and proposing {marriage, fielder} |
| news | + | + | - | Some cosmetics additionally use olive oil as their {base, untrained} |
| nonsense | - | ? | - | ani division good allay tolerant spite despite {stark, crusader} |
| randomseq | - | - | - | Onoclea click bachelorhood cannon fodder Gavialidae polished fatalism baryta {inadmissible, savours} |

Table 4: Overview of datasets used in Exp. 2. "+"and "-" indicate whether a dataset is semantically or syntactically well-formed, or whether the items are included in the training data. The two question marks here: Wikipedia data are very likely to be used to train the models, but this is not certain; Nonsense items have syntactic structures, but not perfectly well-formed. Original final words are shown in blue; randomly picked alternative words are in red.

on unfiltered dataset, as shown in Fig. 7. Since our analysis is conducted on an unfiltered dataset, the differences between different MODELSIMILARITY appear to be smaller, and **self** does not have an advantage. We ran the same regression as in Exp. 1 that predicts $\Delta\text{Meta}_A \sim \Delta\text{Direct}_B$ from $\Delta\text{Direct}_A \sim \Delta\text{Direct}_B$ and MODELSIMILARITY and got similar results: there was a significant effect of $\Delta\text{Direct}_A \sim \Delta\text{Direct}_B$ ($\hat{\beta} = 0.20$, $p < 0.001$), and *higher* values of same family and other were observed (both $ps < 0.01$).

## D  Construction of datasets in Exp. 2

Four datasets were constructed for Exp. 2: wikipedia, news, nonsense and randomseq. For each dataset, we first collected 1000 sentences with 8 to 25 words. The final words of sentences were constrained to be in the top 8k in the BNC/COCA lists[3]. Then, an alternative word that has the same (or closest) lemma frequency in the BNC/COCA lists with the last word was found for each sentence. In this way, we obtained 4000 items, each consisting of a prefix and two different continuations (real and counterfactual).

---

[3] https://www.eapfoundation.com/vocab/general/bnccoca/

**Wikipedia**   To build this dataset, we collected sentences from level 3 vital articles on Wikipedia[4]. As all the articles were published earlier than the knowledge cutoff dates of the models in our study, and the level 3 vital articles are described as most important articles with high quality, it is highly possible that they are in the models' training data.

**News**   To build this dataset, we used NewsData[5] to download English news in October 26-30, 2024 and sampled sentences randomly. As all these news are published after the knowledge cutoff dates of the models[6], they are not expected to be included in the training data. However, the sentences in this dataset should not differ significantly from those in the training data, in terms of overall distribution.

**Nonsense**   To build this dataset, we created nonsense sentences from our **news** dataset. For each sentence, each word is replaced by a word with the same part of speech (we used universal word tags in Brown Corpus (Francis & Kucera, 1979) to label words in BNC/COCA word list). The nonsense sentences generated in this way may not perfectly adhere to grammatical rules (inflectional variations are difficult to handle), but they are generally consistent with the news dataset in terms of syntactic structures.

**Randomseq**   To build this dataset, we generate sequences of 8 to 25 words by randomly selecting words from the BNC/COCA word list, with the final word chosen randomly from the top 8k. These randomly generated sequences are considered out-of-domain for all the LLMs in our study, as they consist of strings lacking any syntactic structure or meaning.

## E   Exp. 2 results by subsets

### E.1   Model preference on answers

Fig. 8 shows the proportion of times the model chooses the original last word rather than the randomly picked alternative word. For the **wikipedia** and **news** datasets, the models overwhelmingly prefer the original word (near 100% across all models), as it is both semantically and syntactically plausible. In contrast, for the **nonsense** and **randomseq** datasets where both continuations are randomly chosen and the sentences are meaningless, the model does not distinguish between the two options when prompted. We attribute the model's preference for the original word in the nonsense dataset to its higher syntactic plausibility: both continuations ($w_1$ and $w_2$ are meaningless, but $w_1$ follows some syntactic rules as its part of speech is the same as the final word in a grammatical sentence. For the **randomseq** dataset, although we controlled lemma frequency when selecting the alternative word, a significant difference in word-form frequency ($p < 0.001$) was observed between the original word and alternative word (average log word form frequency=4.71 for $w_1$ and 5.31 for $w_2$). We believe this difference underlies the model's slight preference for alternative words in the probability measurement.

### E.2   Effect of model size

Similar to Exp. 1, we found that $\Delta$Meta from larger models from larger models tend to more correlated with $\Delta$Direct from all the models across all the datasets, as shown by brighter upper cells in Fig. 9. We ran the same regression predicting mean Pearson $r$ from the log Direct model size, the Meta model size and their interaction. We found that the positive effect of Meta model size is significant ($p<0.001$) across all datasets, the effect of interaction is insignificant and there is a negative effect of the Direct model size on *wikipedia* and *news* datasets.

---

[4]https://en.wikipedia.org/wiki/Wikipedia:Vital_articles/Level/3

[5]https://newsdata.io/

[6]The training data cutoff date is not released for Qwen and Mistral models, but it is not likely for them to be trained on data after Oct 2024, as the models are released on Sep 2024 (Qwen) and Nov 2024 (Mistral) respectively.

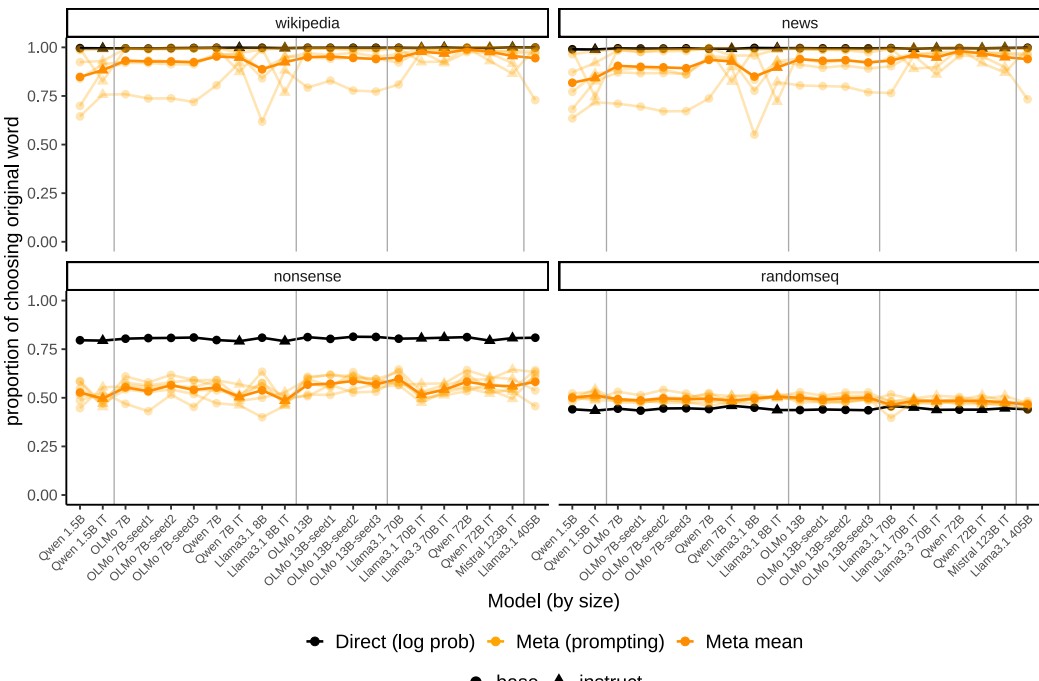

Figure 8: The proportion of choosing answer #1 (original last word in the sentence) of the models in the four datasets. Each yellow line stands for a specific prompt.

## F Analysis on seed variants of the OLMo models

Fig. 10 shows a specific breakout for studying the OLMo models, the only models for which we have models that are identical besides their random seeds. We examined the within- and cross-model consistency of the 7B and 13B OLMo models and their variants. The three *B – seed* models for each parameter size are trained on the same data but with a different data order (different random seeds used), and the models "OLMo 7B" and "OLMo 13B" are final models with averaged weights. In these models trained on same data, there is still no sign of $\Delta \text{Meta}_A \sim \Delta \text{Direct}_A > \Delta \text{Meta}_A \sim \Delta \text{Direct}_B$. This pattern serves as strong support for our conclusion, that although $\Delta \text{Meta} \sim \Delta \text{Direct}$ is a function of MODELSIMILARITY, this does not indicate the introspection on knowledge of language. We ran regression models predicting $\Delta \text{Meta}_A \sim \Delta \text{Direct}_B$ with 'whether the pair is **self**' to examine the effect. Across all the datasets (filtered and unfiltered in Exp. 1, as well as four datasets in Exp. 2), no significant effect of **self** was observed (all $ps > .25$).

## G Regression models

In this section, we present the details and implementation of the regression models.

**Predict correctness from MODELSIZE and method** For predicting model correctness from MODELSIZE and method (§3.2.1), we ran a logistic regression:

```
glm(value ~ DirectOrMeta * log(model size), family = 'binomial')
```

The dependent variable `value` is a Boolean value indicating whether the model answered *each question* correctly. `model_size` refers to the size of the model (regardless of whether Direct or Meta). `method` refers to whether it is Direct or Meta.

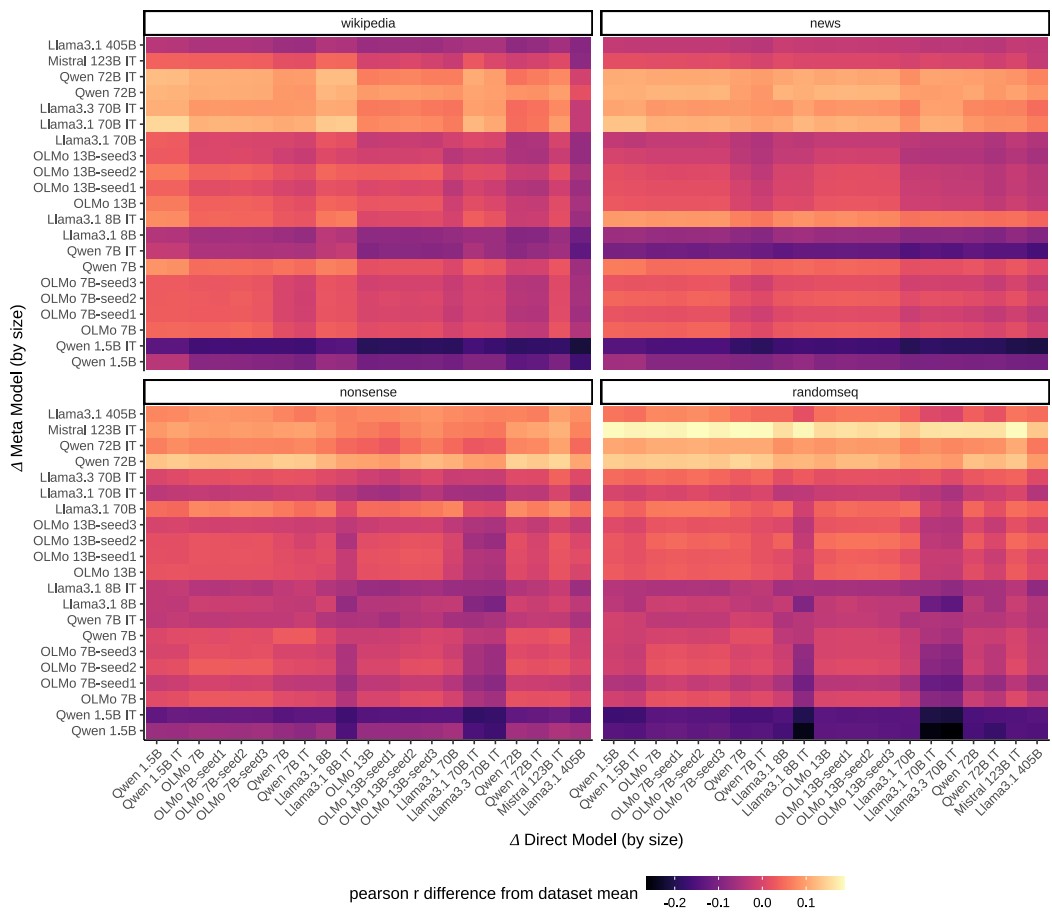

Figure 9: Pearson *r* (averaged across prompts) for each pair of models on four datasets in Exp. 2.

|  | estimation |
|---|---|
| Intercept | 1.6436*** |
| IsMeta | −0.5758*** |
| log model size | 0.0305*** |
| Meta:log model size | 0.2211*** |
| $^{***}p < 0.001; ^{**}p < 0.01; ^{*}p < 0.05$ | |

Table 5: $\hat{\beta}$ coefficients and p-values for logistic regression predicting correctness from Direct/Meta and log model size.

**Predict $\Delta\text{Meta}_A \sim \Delta\text{Direct}_B$ Pearson *r* from log MODELSIZE**   In addition to using heatmaps to visualize the $\Delta\text{Meta}_A \sim \Delta\text{Direct}_B$ (Pearson *r*) between each pair of models (Figure 2b, Fig. 6, Fig. 9), we ran the following regression predicting $\Delta\text{Meta} \sim \Delta\text{Direct}$ with model size:

```
lm(value ~ log(Meta model size) * log(Direct model size))
```

Where `mean_prompt_corr` stands for $\Delta\text{Meta}_A \sim \Delta\text{Direct}_B$ Pearson *r* (averaged across prompts), `prompt_size` stands for the size of the $\Delta\text{Meta}$ model and `prob_size` stands for the size of the $\Delta\text{Direct}$ model. Table 6 presents the regression models run on the results obtained from different datasets in this experiment.

| | exp1(filtered) | exp1(unfiltered) | wikipedia | news | nonsense | randomseq |
|---|---|---|---|---|---|---|
| Intercept | 0.1014*** | 0.3191*** | 0.3143*** | 0.3361*** | 0.0720*** | 0.0603*** |
| log Direct model size | −0.0040 | 0.0025 | −0.0161*** | −0.0119* | −0.0057 | −0.0063 |
| log Meta model size | −0.0012 | 0.0135*** | 0.0233*** | 0.0230*** | 0.0302*** | 0.0432*** |
| Interaction | 0.0035*** | 0.0013 | 0.0004 | 0.0014 | 0.0007 | −0.0001 |

$^{***}p < 0.001; ^{**}p < 0.01; ^{*}p < 0.05$

Table 6: $\hat{\beta}$ coefficients and p-values for regression predicting ΔDirect $\sim$ ΔMeta from log model size for the Direct model and log model size for the Meta model.

**Predict ΔMeta $\sim$ ΔDirect Pearson $r$ from MODELSIMILARITY and ΔDirect $\sim$ ΔDirect** In addition to visualizing the relationship between ΔMeta$_A$ $\sim$ ΔDirect$_B$ (Pearson $r$) and MODELSIMILARITY (both categorical and empirical) in Fig. 3, Fig. 4 and Fig. 7, we ran the following regression predicting ΔMeta $\sim$ ΔDirect with both ΔDirect $\sim$ ΔDirect and categorical MODELSIMILARITY:

```
lm(value ~ DirectDirect + ModelSimilarity)
```

Here, we use **self** as baseline and see the effects of other MODELSIMILARITY (**seed variant**, **base/instruct**, **same family**, **other**). Table 7 presents the regression models run on the results obtained from different datasets in this experiment.

| | exp1(filtered) | exp1(unfiltered) | wikipedia | news | nonsense | randomseq |
|---|---|---|---|---|---|---|
| Intercept | 0.0322 | 0.1832*** | −0.0428 | −0.4439* | −0.3315*** | −0.3849*** |
| ΔDirect $\sim$ ΔDirect | 0.1026*** | 0.1998*** | 0.3959*** | 0.8338*** | 0.4913*** | 0.5613*** |
| seed variant | 0.0043 | −0.0168 | 0.0116 | 0.0130 | 0.0121 | 0.0198 |
| base/instruct | 0.0203 | 0.0256 | 0.0068 | 0.0055 | 0.0146 | 0.0020 |
| same family | 0.0519*** | 0.0404** | 0.0483* | 0.0550* | 0.0651*** | 0.0650** |
| other | 0.0504** | 0.0501** | 0.0479** | 0.0699** | 0.0791*** | 0.0887*** |

$^{***}p < 0.001; ^{**}p < 0.01; ^{*}p < 0.05$

(a)

| | exp1(filtered) | exp1(unfiltered) | wikipedia | news | nonsense | randomseq |
|---|---|---|---|---|---|---|
| Intercept | 0.1406** | 0.3854* | −0.2688 | −0.1654 | −0.0526 | −0.0454 |
| ΔDirect $\sim$ ΔDirect | 0.0294 | 0.0367 | 0.6456*** | 0.5926 | 0.2571 | 0.2754 |
| base/instruct | 0.0119 | 0.0046 | 0.0606 | 0.0303 | 0.0110 | 0.0027 |
| same family | −0.0208 | 0.0192 | 0.0727* | 0.0135 | −0.0019 | −0.0249 |
| other | −0.0152 | −0.0090 | 0.1039** | 0.0469 | 0.0499 | 0.0550 |

$^{***}p < 0.001; ^{**}p < 0.01; ^{*}p < 0.05$

(b)

Table 7: $\hat{\beta}$ coefficients and p-values for regression predicting ΔDirect $\sim$ ΔMeta from ΔDirect $\sim$ ΔDirect and MODELSIMILARITY using data from (a) all models (b) models larger than 70B.

Table 7a shows on t the results we present in the main text, whereas Table 7b shows results where we focus in on just comparisons among "big models", defined as those of 70B parameters or greater. As in the main text analysis, we would take a significant *negative* value for one of the MODELSIMILARITY coefficients to suggest an effect of introspection, but we do not find that. So, even among big models, we do not see evidence of introspection.

**Predict ΔMeta $\sim$ ΔDirect Pearson $r$ in OLMo models** We did a more fine-grained analysis of only the 7B and 13B OLMo, because they allow comparing models that are identical besides the random seed. We find no evidence of introspection. As there are only three types of MODELSIMILARITY among those models (**self**, **seed variant** and **same family** for 7B / 13B models), we focused on the effect of **self** and ran a regression predicting ΔMeta$_A$ $\sim$ ΔDirect$_B$ Pearson $r$ by whether $A = B$ (is **self**):

$$\text{lm(value} \sim \text{IsSelf)}$$

As shown in Table 8, we do not observe any evidence of introspection among the OLMo models, as there is no significant effect of **self** across all datasets.

|  | exp1(filtered) | exp1(unfiltered) | wikipedia | news | nonsense | randomseq |
|---|---|---|---|---|---|---|
| Intercept | 0.1446*** | 0.3621*** | 0.3540*** | 0.3952*** | 0.1547*** | 0.1799*** |
| **self** | −0.0106 | 0.0007 | 0.0071 | 0.0031 | 0.0060 | 0.0080 |

$^{***}p < 0.001; ^{**}p < 0.01; ^{*}p < 0.05$

Table 8: $\hat{\beta}$ coefficients and p-values for regression predicting effect of **self** in OLMo models (with a targeted focus on **seed variant** and **same family** 7B and 13B models comparison).

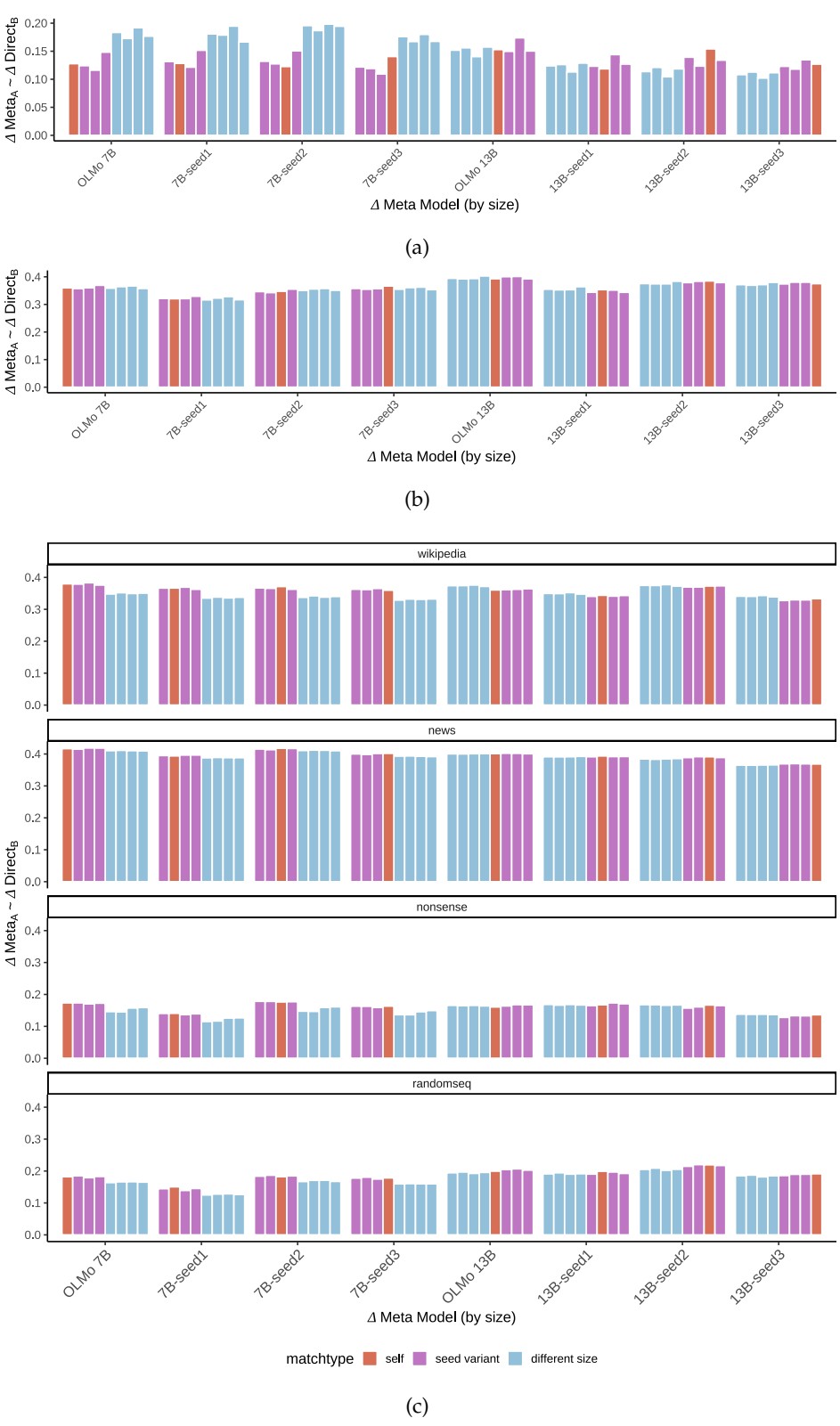

Figure 10: Pearson *r* for the OLMo models in (a) Exp. 1 with filtered dataset (b) Exp. 1 with unfiltered dataset (c) Exp. 2. There is no clear trend for higher correlation for the same model, compared to its seed siblings and other models in the same family.

