# OpenReview forum: "Language Models Fail to Introspect About Their Knowledge of Language"
_colmweb.org/COLM/2025/Conference — COLM 2025_

### Official Review · Reviewer_n8wV · 2025-05-07

**Rating:** 4
**Confidence:** 5
**Ethics Flag:** 1

**Summary:**

Significance:
- This paper tackles an interesting question of whether models have privileged "Self-access" to their internal states. If so, this could help e.g. interpretability of language models.
- The paper finds instead that there is no evidence for this, in language models not fine-tuned for introspection.


Clarity:
- I found this paper difficult to read. One problem is that Figure 1 contains 4 different figures that are referenced across the paper. It is different to understand Figure 1 as a whole. When the other parts of the paper refer to Figure 1, it is not clear which specific figure is being referred to. For example, one of the later pages references a part of the Figure 1 as "Figure 1c, bottom". This is difficult for readers to locate.
- Suggestion: I think the readability of the paper would be much improved if the authors would break up Figure 1 into separate figures, and introduced them alongside the relevant parts of the paper. Add further elaboration on the respective broken down parts of the figure in the captions.


Originality:
- The paper is original in investigating further the question of self-access. Not much work has been done in this area. The paper claims to complicate recent results suggesting that models can introspect.
- Still, I question whether this paper makes an original contribution, and whether it complicates prior work. (Elaborated later)


Quality:
- The authors conduct experiments across 21 different models to check for introspection. They conduct two main experiments. One with introspection about grammar, and another with introspection about next word prediction.
- Due to the questions I have about originality, and the questions about the execution of the experiments, I currently do not find the paper to be of high quality.

**Reasons To Accept:**

This submission examines multiple models to check for introspection. The claim that "If models are not finetuned for introspection, they will fail to introspect" seems substantiated by their two experiments.

**Reasons To Reject:**

### On contribution and whether it complicates prior work
I question whether this submission makes an original contribution and whether it complicates prior work.
- This submission investigates a scenario where models are not fine-tuned for introspective tasks. It then concludes that there is no evidence for introspection in these tasks. The authors say that this result challenges prior work, such as Binder et al. (2024).
- But, examining Binder et al. (2024), they already state the same conclusion when models are not fine-tuned. In Binder et al. (2024), they show that models perform at merely baseline levels unless they are fine-tuned for introspection (Binder et al., 2024, Figure 4).
- This submission acknowledges this fact by saying they do not conduct fine-tuning experiments, unlike Binder et al.
- If so, then what exactly is the contribution of this paper? This submission only investigates scenarios without fine-tuning, which prior work cited has already shown to not have evidence for introspection.
- Please clarify -- What prior work suggests that introspection would work without dedicated fine-tuning? Pointing out an exact paper to look at would be good. How does this submission "complicate" results?
- I suggest the authors clarify their exact contribution.


I can think of two ways forward:
- Refute my claim about what the cited work shows. Show that the cited or other prior work infact claims that models can introspect without dedicated fine-tuning.
- Do fine-tuning for introspection using your setup. Show that there is still no self-access advantage. Binder et al. claims that a large model Llama 70b can introspect, so maybe you can compare directly there? Show that even though you are comparing the same model, you get different results.

Note: They also speculate that weaker models like GPT-3.5 fail to introspect,  so introspection on small models will be expected to fail (Binder et al., Figure 4). One problem is that there are no multiple seeds for Llama 70b, but only for small models 7,13B for OLMo-2.


### On methodology
The authors state that they "focused on the items for which ≥5% of the models disagree under the Direct
196 and Meta evaluations. We did this because there is limited potential to observe introspection
197 for items which are so easy that all models and methods provide the same answer.".

I agree with this premise. However, just greater than 5% is not a very stringent threshold. Doesn't that mean that they require only 2 models out of the 21 models to disagree on a certain sample for it to be considered for analysis? Perhaps, when comparing two models A and B, it would be better to instead filter only for examples where the models A and B disagree?
This filtering follows prior work (Kadavath et al., 2022, Figure 20, Page 21), which looks at cross-experiments on similar models A and B.

Kadavath et al. (2022).  Language Models (Mostly) Know What They Know. https://arxiv.org/abs/2207.05221

---

> ### Author Response · Authors · 2025-05-30
>
> We very much appreciate this thorough review, which raises important points to address.
>
> Re: relationship to Binder et al: To provide a bit more context, we started this work last summer before Binder et al. (2025) was preprinted, and we learned about their work during the writing of our paper. So our design decisions were not made with any awareness of Binder et al.. We tried to contextualize our results with respect to what we took as the main takeaways from Binder et al., but we agree we can do better. We took the main point of Binder et al. to be that introspection emerges with fine-tuning, but we agree we did not contextualize this fully in light of the negative results for non-fine-tuning in that paper, which we didn’t fully appreciate. In the revision, we will clarify how our results relate to those of Binder et al., and exactly what we mean by “complicating” prior work.
>
> That said, because we came at the topic of introspection independently of Binder et al., we think there are several independent contributions in our work:
> 1. We think there is value in having a related but different way of operationalizing introspection.
>    a. As we justify in the paper, we think that there is good reason to use the method we use of “controlling for model similarity,” specifically using a method that taxonomizes different kinds of model similarity.
>    b. A priori, we think it is reasonable to adopt a definition of “introspection” as an ability which emerges without explicit fine-tuning. This is how introspection is typically treated in cognitive science and linguistics: there is every effort made to obtain intuitions about things like grammaticality without giving explicit instructions (see Sprouse, Schütze, Almeida, 2013; or Schütze, 1996; for some of the argumentation and empirical basis of this method). Furthermore, there is a question of whether a fine-tuned version of a model predicting the pre-fine-tuned version can truly be thought to be predicting “itself,” since its parameters have changed. To be clear, we do not mean this to criticize the approach taken by Binder et al., which we think is methodologically sound. We simply think it is worth testing the non-fine-tuned setting in more detail (across a range of models, model sizes, model families, tasks, etc.), even if it has been shown to not work by Binder et al.
>
> 2. We think there is value in having a different set of tasks and evaluation methods.
>    a. We in particular think our nonsense and random sequence tasks provide interesting test conditions going beyond those in other introspection work.
>    b. Our evaluation requires mapping between log probabilities and a metalinguistic prompt. This is somewhat unique to the linguistic domains we focus on, but we think is particularly useful for studying introspection since it doesn’t require “prompt to prompt” mapping but rather “prompt to direct probability space.” The probabilities that LMs assign to strings are the most direct form of “knowledge” that we can measure, since that is exactly what they are trained to provide.
>
> 3. Part of our motivation for this work was from a linguistics angle. In linguistics, acceptability judgments are a key way of studying models in humans. They are taken to reflect humans' grammatical knowledge. The issue of whether LMs can “introspect” remains hotly debated (e.g., Dentella et al. 2023, Hu et al. 2024), and would have important implications for linguistics research and theory. We would like to, in the Camera Ready, further draw out the implications of this work for linguistics.
>
>
> We agree it would be interesting to see if these results improve with fine-tuning, similar to in Binder et al. This is something we could aim to do either for the Camera ready or a follow up. But our method is not straightforwardly amenable to fine-tuning since we’d have to fine tune to predict log probability differences. This would require further exploration to see if it can be done in a reasonable way, with our compute resources.
>
> Overall, we believe, even in light of Binder et al., that there is more work needed in the model introspection space. And we are glad the reviewer agrees in part with that: “The paper is original in investigating further the question of self-access. Not much work has been done in this area.”, even while recognizing that we should better situate our work relative to Binder et al.’s important study.
>
> Re: figure: Yes, we can use the extra space to explore breaking the figure up into parts, for clarity.
>
> Re: methodology: Yes, 5% means that just 1 disagreement has to occur. We tried varying this threshold quite a lot and did not find major differences and so did not focus on that. We do think the results are robust to this choice of threshold. Thanks for the pointer to Kadavath et al. (2022) which is useful for having a precedent.
>
> Thank you again for this very helpful and thoughtful review.

---

> > ### Comment · Reviewer_n8wV · 2025-06-08
> >
> > Thank you, I now write this followup on what we have agreed so far.
> >
> > You have investigated introspection in terms of string probability.
> > We find that current models dont have introspection in terms of the definition you have outlined.
> >
> > In terms of impact, I do not find that this is a big update in terms of the fields knowledge. Similar systems of string probability have been shown ( as you have cited in the paper ).
> >
> > In terms of measuring it in the latest models, this is still useful, and I give you credit for it.
> > You show that introspection does not exist in your definition.
> >
> > But, previous work has already shown this to be true (as we have discussed).
> >
> > So your contribution is showing further evidence that there is no introspection without finetuning. I think this is not a big update in terms of what the field already knows.
> > I still give you credit in terms of showing this evidence, so I have raised my score marginally.

---

> ### Comment · Reviewer_n8wV · 2025-06-01
>
> Thanks for the discussion.
> >We think there is value in having a related but different way of operationalizing introspection ...
>
> Yes, I think this discsussion would be good to add into the paper. How exactly is your method different and what exactly does it complicate prior work? I think readers would find it helpful. Otherwise it would be confusing to readers (as I was) about the claims this paper makes.

---

> > ### Author Response · Authors · 2025-06-02
> >
> > Thanks for this response. We definitely plan to update the paper to highlight the main takeaways we brought out in this discussion. And we also will update this line in the abstract: "Our findings complicate recent results suggesting that models can introspect" to focus on the methodological and task innovations of the paper.
> >
> > We think these changes, which would not require redoing any experiments, results, or analyses, could be comfortably handled in a revision for the Camera Ready and would make the paper stronger. If you agree, we would be happy for you to reconsider the assessment of "Clear Reject" (while of course respecting your prerogative to maintain it if you think that's warranted). Or we are happy to discuss further here.

---

### Official Review · Reviewer_PXdC · 2025-05-13

**Rating:** 7
**Confidence:** 2
**Ethics Flag:** 1

**Summary:**

This work revisits the debate about whether LLMs can demonstrate awareness of their own linguistic knowledge.

The experimental methodology centers on the observation that a model's internal linguistic knowledge can be grounded in computing the appropriate string probability. A model's preference can be either computed directly or elicited via a meta-linguistics prompt. Then, pairs of models can be compared by seeing how they align in the difference between the direct and meta-linguistics measurements. The experimental set up aims to overcome some of the limitations in earlier studies by explicitly controlling for similarities between models.

**Reasons To Accept:**

The experiment is cleverly designed. While the evaluation is limited to grammatical correctness and sentence completion, the domain choices made it possible to know the ground truth by computing string probabilities. This way, the work is able to compare many models (21 open-source LLMs) in a controlled fashion.

The results of the study are interesting and thought-provoking. Under the proposed experimental methodology, the results do not provide evidences that support model introspection, which contradicts some prior work.

**Reasons To Reject:**

As the authors pointed out themselves -- that their proposed methodology did not provide evidence for introspection does not necessarily mean that a model cannot introspect.

The notion of introspection as determined through comparing "direct" and "meta" is not very intuitive for people who are not used to thinking about this problem. It would be nice to have a stronger motivation for why the "meta" method is accessing the same internal information as "direct."

---

> ### Author Response · Authors · 2025-05-30
>
> Thanks for this positive assessment! Yes, we agree that negative evidence requires careful conclusions. Another reviewer also found “direct” and “meta” unintuitive. We will try to clarify in a figure to make sure the terms are clear.

---

> > ### Comment · Reviewer_PXdC · 2025-06-05
> > **Acknowledged**
> >
> > Thanks for the comment. I look forward to the next iteration of this paper.

---

> ### Comment · Area_Chair_RYiq · 2025-06-03
> **Could you engage with or just acknowledge the authors' response?**
>
> Thank you!

---

### Official Review · Reviewer_KaWq · 2025-05-13

**Rating:** 8
**Confidence:** 3
**Ethics Flag:** 1

**Summary:**

This paper addresses a question of whether large language models (LLMs) possess the capacity for introspection—that is, whether they can reliably reflect on their own internal linguistic representations. The authors examine this question across 21 open-source LLMs, focusing on two domains of theoretical interest: grammaticality judgments and word prediction. In both cases, the models’ underlying linguistic knowledge is operationalized through direct measurements of string probability (direct method), providing a grounded basis for comparison. Additionally, a meta method compares the log probabilities assigned to responses to metalinguistic prompts.

The study proposes a metric for introspective ability as the degree to which the model's prompted responses predict its own string probabilities, compared to what would be predicted by another model with nearly identical internal knowledge. The method was evaluated on 670 minimal pairs from BLiMP and 378 minimal pairs from Linguistic Inquiry for grammaticality judgement and 4 datasets with 1000 items each for the word prediction.  However, because the introspective ability metric depends on the difference between one model’s response and that of a highly similar model, a much smaller subset of minimal pairs was eventually analyzed for the grammaticality judgement, where over 5% model disagreement yielded only 294 minimal pairs.

Although both direct and meta approaches yield high task accuracy, the authors do not find evidence that LLMs have privileged self-access to their internal states.

**Questions To Authors:**

Model agreement resulted in a substantial reduction in the number of minimal pairs. Could further analysis reveal patterns in the types of data on which the models tended to agree and whether similar types occurred in the data analyzed for introspection?

**Reasons To Accept:**

This is a well-written paper with a clear methodological approach, a solid theoretical foundation, and well-articulated results. The significance of the findings is thoughtfully discussed, raising important questions about whether language models genuinely access their internal states and to what extent prompted responses reflect underlying linguistic knowledge.

**Reasons To Reject:**

no major reasons found

---

> ### Author Response · Authors · 2025-05-30
>
> Thanks for this positive assessment! We like this idea of testing further what kinds of data generates high agreement: this is also potentially linguistically interesting if there is linguistic structure to it. We will aim to do an analysis like this either for the CR or for a follow-up.

---

> > ### Comment · Reviewer_KaWq · 2025-06-06
> >
> > Thank you for the comment. I look forward to reading the analysis!

---

> ### Comment · Area_Chair_RYiq · 2025-06-03
> **Could you engage with or just acknowledge the authors' response?**
>
> Thank you!

---

### Official Review · Reviewer_gFfK · 2025-05-13

**Rating:** 8
**Confidence:** 4
**Ethics Flag:** 1

**Summary:**

This paper investigates the degree to which LLMs are capable of introspection. In contrast to the work of Binder et al. (2025), in this paper, the conclusion based on empirical results is that the authors find no evidence of introspection or other privileged access to self-information. This finding holds across 21 open-source LLMs in two domains.

Introspection is operationalized as the degree to which a model’s metalinguistic (prompt-based) responses are correlated with its literal token probabilities beyond the correlation that can be inferred by another model with approximately the same internal knowledge. The premise of the paper is that (line 190) “if model A introspects, then we should find ∆Meta_A ∼ ∆Direct_A > ∆Meta_A ∼ ∆Direct_B for any other model B, even if A and B are highly similar” where “∆Meta_A ∼ ∆Direct_A” is the correlation between metalinguistic and literal model inferences of model A, and similarly “∆Meta_A ∼ ∆Direct_B” between the metalinguistic knowledge of model A versus the literal probabilities of model B.

Because ∆Meta_A and ∆Direct_B are correlated with the similarity of model A and B (under two different definitions of similarity) BUT not more than can be predicted based on similarity alone, the authors conclude no evidence of introspection. E.g. two models  that are nearly the same (e.g. two instances of OLMo with different initial random seeds before pretraining) have essentially the same predictive power of each other’s literal token probabilities as they do of their own based on metalinguistic prompting.

**Questions To Authors:**

Questions:
1. How can we know whether the final conclusion is due to no introspection versus due to lack of discriminative power of the experiment?
2. Because the empirical definition of similarity is measuring the exact attribute that the paper is testing whether or not a model is capable of introspecting about (literal token probabilities), would we apriori expect a nearly identical model ∆Meta_A ∼ ∆Direct_A = ∆Meta_A ∼ ∆Direct_B?

**Reasons To Accept:**

- The paper is incredibly clearly written.
- The methodology appears sound and there are extensive supporting experiments.
- While one could disagree with the operationalization of introspection, I find it overall reasonable and supported by the experiments.

**Reasons To Reject:**

- The biggest weakness of the paper is possibly the narrow operationalization of introspection.That being said, I see how this operationalization allows for evaluation, and the paper includes discussion of why conclusions might differ from Binder et al.
- It was not clear to me at what ∆Meta_A ∼ ∆Direct_A the “same model effect” would be considered to take place. E.g. in Figure 1d, the rightmost point still appears to correspond to a higher value on the y-axis than distinct but similar models. Is this not considered the same model effect because the increase is sub-linear?
- I was surprised there is not engagement in earlier work studying “explainability” or “truthfulness” of models. E.g. [1]

[1] A Survey of the State of Explainable AI for Natural Language Processing (Danilevsky et al., AACL 2020)

---

> ### Author Response · Authors · 2025-05-30
>
> Thank you for your helpful comments and suggestions.
>
> Re: “narrow operationalization of introspection”: Thank you for pointing this out. Our operationalization of introspection (i.e., metalinguistic responses predicting literal token probabilities) is directly inspired by the way that introspection is often used in linguistics and cognitive science. That is: using a high-level metacognitive question to gain information about an internal state. We agree that there could certainly be alternate ways to operationalize introspection, e.g. the fine-tuning approach taken by Binder et al. 2025, which would provide complementary insights to our approach. (See response to n8wV for more on this.)
>
> Re: “same model effect”: Yes —in short, if there is a ‘same model effect’, then within-model correlation should be higher than what would be predicted on the basis of model similarity. Our linear regression analyses showed that the within-model correlation results are indeed predicted by model similarity, but there was no evidence for the ‘same model effect’ which would be suggestive of introspection. (lines 262-265, 323-327, Appendix G)
>
> Re: engaging with “explainability and truthfulness”: Thank you for suggesting these references! We will add discussion of this literature in the extra space.
>
> Re: discriminative power of the experiments: Thank you for this important question. As with any null result, it is certainly possible that we might have found an effect of introspection in a different set of models/tasks/etc. (as we wrote in the Discussion). However, we designed our experiments to give models “a fair shot” at demonstrating introspection whenever possible. For example, in Experiment 1, we only analyzed the “hard” items where there was meaningful disagreement across models, which would make it easier to observe introspection, a priori. In Experiment 2, we used out-of-domain stimuli such as nonsense sentences and random word sequences, to also make the conditions for observing introspection as favorable as possible. In addition, we made our best effort to cover a wide range of model families and sizes (1.5B to 405B). So, while it is possible that introspection would be observed in some other setting, we feel that our experiments are designed with reasonable discriminative power.
>
> Re: empirical definition of similarity: Yes, that is correct — for two nearly identical models A and B, we would expect ∆Direct_A to be strongly correlated with ∆Direct_B, and so a priori we would also expect ∆Meta_A ∼ ∆Direct_A to be similar to ∆Meta_A ∼ ∆Direct_B. However, there is still room for ∆Meta_A ∼ ∆Direct_A to be stronger than ∆Meta_A ∼ ∆Direct_B, which would be suggestive of a “same model effect”. Empirically we do not observe this in our experiments. Of course, the more similar A and B are, the harder it is to observe this effect, which highlights the importance of testing models which range in similarity to each other.

---

> > ### Comment · Reviewer_gFfK · 2025-06-05
> >
> > Thank you for these clarifications.
> >
> > Wr.t. disagreement with Reviewer n8wV, I agree with the authors' perspective that "A priori, we think it is reasonable to adopt a definition of “introspection” as an ability which emerges without explicit fine-tuning".
> >
> > While I agree finetuning experiments could possibly be interesting, I disagree that finetuning experiments are needed in a future camera-ready version of this work. Rather, I believe such experiments would be more appropriate in a future follow up, so as not to distract from the perspective taken in the current work under review. (Specifically in response to the authors' comment "We agree it would be interesting to see if these results improve with fine-tuning, similar to in Binder et al. This is something we could aim to do either for the Camera ready or a follow up.)

---

> ### Comment · Area_Chair_RYiq · 2025-06-03
> **Could you engage with or just acknowledge the authors' response?**
>
> Thank you!

---

### Decision · Program_Chairs · 2025-07-08

**Decision:**

Accept

**Comment:**

Most of the reviewers saw merit in this paper, and I agree. The experiments are empirically sound, the results interesting, and I appreciated the controls for model similarity. I do have a few concerns with the framing, which I encourage the authors to take into consideration for the camera ready if this paper is accepted. First, the idea that *humans* have reliable access to their own linguistic knowledge when they provide an acceptability judgment or complete a sentence in the cloze task, without any interference from task demands, seems simplistic. How strongly can be expect the implicit and explicit measures to be correlated in LLMs? Second, the paper sometimes appears to be about introspection more generally (meta-cognition), but in fact is focused on two particular aspects of the knowledge of language. Third, the contrast with Binder et al. (2025) is not sufficiently clear; related to the previous point, it seems perfectly plausible for models to have metacognitive access to some computations but not others. Because Binder's work is so prominent in the framing, I’d expect some more details on what that paper found and how inconsistent that is with the current results. The discussion with reviewer n8wV includes some useful ideas for addressing this issue. Finally, I wonder a more direct test of introspection wouldn’t be to simply ask the model something like “the probability of sentence 1 is higher than the probability of sentence 2” and compare THAT to the difference in probability assigned to those sentences by the model; as it currently stands, this test relies on the targeted syntactic evaluation paradigm providing a direct window into the model’s understanding of acceptability, which adds another assumption to the test. In general I encourage the authors to keep in mind that COLM has a broad range of audiences, so terminology, methods and approaches that are common in linguistics may need to be presented more gently.